

# Land snails of *Leptopoma* Pfeiffer, 1847 in Sabah, Northern Borneo (Caenogastropoda: Cyclophoridae): an analysis of molecular phylogeny and geographical variations in shell form

Chee-Chean Phung, Pooi-San Heng and Thor-Seng Liew

Institute for Tropical Biology and Conservation, Universiti Malaysia Sabah, Kota Kinabalu, Sabah, Malaysia

## ABSTRACT

*Leptopoma* is a species rich genus with approximately 100 species documented. Species-level identification in this group has been based on shell morphology and colouration, as well as some anatomical features based on small sample sizes. However, the implications of the inter- and intra-species variations in shell form to the taxonomy of *Leptopoma* species and the congruency of its current shell based taxonomy with its molecular phylogeny are still unclear. There are four *Leptopoma* species found in Sabah, Borneo, and their taxonomy status remains uncertain due to substantial variation in shell forms. This study focuses on the phylogenetic relationships and geographical variation in shell form of three *Leptopoma* species from Sabah. The phylogenetic relationship of these species was first estimated by performing Maximum Likelihood and Bayesian analysis based on mitochondrial genes (16S rDNA and COI) and nuclear gene (ITS-1). Then, a total of six quantitative shell characters (i.e., shell height, shell width, aperture height, aperture width, shell spire height, and ratio of shell height to width) and three qualitative shell characters (i.e., shell colour patterns, spiral ridges, and dark apertural band) of the specimens were mapped across the phylogenetic tree and tested for phylogenetic signals. Data on shell characters of *Leptopoma sericatum* and *Leptopoma pellucidum* from two different locations (i.e., Balambangan Island and Kinabatangan) where both species occurred sympatrically were then obtained to examine the geographical variations in shell form. The molecular phylogenetic analyses suggested that each of the three *Leptopoma* species was monophyletic and indicated congruence with only one of the shell characters (i.e., shell spiral ridges) in the current morphological-based classification. Although the geographical variation analyses suggested some of the shell characters indicating inter-species differences between the two *Leptopoma* species, these also pointed to intra-species differences between populations from different locations. This study on *Leptopoma* species is based on small sample size and the findings appear only applicable to *Leptopoma* species in Sabah. Nevertheless, we anticipate this study to be a starting point for more detailed investigations to include the other still little-known (*ca.* 100) *Leptopoma* species and highlights a need to assess variations in shell characters before they could be used in species classification.

Corresponding author
Thor-Seng Liew,
thorsengliew@gmail.com

## INTRODUCTION

The terrestrial snail genus *Leptopoma* is one of 35 genera in the family Cyclophoridae (*Kobelt, 1902*), which has a wide global distribution range extending across much of the Oriental and Australasia zoogeographical regions. An early global overview of species-level diversity in *Leptopoma* species classified the genus into four subgenera with a total of 105 species (*Kobelt, 1902*). Several subsequent regional taxonomic reviews of *Leptopoma* were conducted for the Philippines (*Zilch, 1954*), South Asia (*Gude, 1921*), and most recently for Borneo (*Vermeulen, 1999*). To date, taxonomic works on *Leptopoma* (*Kobelt, 1902*; *Gude, 1921*; *Zilch, 1954*; *Vermeulen, 1999*) have been based mainly on shell morphology (i.e., shell size, shape, colour pattern and sculpture). Besides shell morphology, other characters of the soft body such as radula, operculum, and genital duct have been used from time to time in the species delimitation (*Sarasin & Sarasin, 1899*; *Jonges, 1980*). Although several species have been included in phylogenetic studies focussing on other taxa (*Colgan, Ponder & Eggler, 2000*; *Colgan et al., 2003*; *Colgan et al., 2007*; *Lee, Lue & Wu, 2008a*; *Lee, Lue & Wu, 2008b*; *Nantarat et al., 2014a*), little is known about the relationship within the genus *Leptopoma* itself.

The genus *Leptopoma* is widespread in the Philippines and the adjacent Malaysian state of Sabah, which is located in the northern part of Borneo (*Godwin-Austen, 1891*; *Laidlaw, 1937*; *Vermeulen, 1999*; *Schilthuizen & Rutjes, 2001*; *Uchidal et al., 2013*). Currently, four *Leptopoma* species could be identified from the specimens collected in Sabah. Of these, *Leptopoma undatum* (*Metcalfe, 1851*) and *Leptopoma atricapillum* (Sowerby, 1843) can be found in Borneo and the Philippines (*Vermeulen, 1999*). *Leptopoma sericatum* (*Pfeiffer, 1851*) is distributed in Borneo (*Vermeulen, 1999*). *Leptopoma pellucidum* (*Grateloup, 1840*) is widely spread in Sabah but the actual global range is unknown due to taxonomy uncertainty (*Vermeulen, 1999*). Its putative synonym, *L. vitreum* (*Duperrey, Lesson & Garnot, 1830*), has a wide range encompassing Taiwan, South Asia, and Papua New Guinea (*Vermeulen, 1999*).

*Leptopoma undatum* is readily distinguished from the others by its uniform white shell (translucent when young and opaque when old) and distinctive shell shape (i.e., relatively less convex whorls and sharply keeled at the last whorl). The other three species, *Leptopoma atricapillum*, *L. sericatum* and *L. pellucidum*, are very similar in terms of shell shape with all showing and sharing colour pattern polymorphism. *L. atricapillum* and *L. sericatum*, however have strongly-defined spiral ridges on the shell surface, with more pronounced spiral ridges in the former, whereas the spiral ridges of *L. pellucidum* are only weakly defined.

The Cyclophoridae represents the most diverse Caenogastropoda family, but is remains poorly resolved taxonomically. Delimitation among subgenera and species in Cyclophoridae has long been a conundrum for taxonomists due to exceptionally diverse variation in morphology (e.g., subgenera in genus *Cyclophorus* and *Alycaeus* (*Kobelt, 1902*; *Gude, 1921*)). *Vermeulen (1999)* identified two major challenges with using shell characters to discriminate between six species of Bornean *Leptopoma*. First, the majority of species are similar in shell morphology and this limits the number of shell characters that can be used as diagnostic characters at species level. Second, there appears to be a continuum
of variation in some characters (i.e., size, shape, colour patterns), particularly between *L. pellucidum* and *L. sericatum* and this contributes to uncertainty when delimiting species. Thus it is clear that to date, the implications of the intra- and inter-specific variation in shell morphology have not been studied systematically and comprehensively in the context of the taxonomy of this genus.

This study had three aims: (1) to estimate the molecular phylogenetic relationship of three similar yet polymorphic *Leptopoma* species in Sabah to investigate the monophyly of *L. sericatum*, *L. pellucidum* and *L. atricapillum* based on two mitochondrial genes (16S rDNA and COI) and nuclear gene (ITS-1); (2) to test the phylogenetic signal of three qualitative and six quantitative shell characters to evaluate their reliability as diagnostic characters; (3) to compare the differences in shell characters of two *Leptopoma* species, *L. sericatum* and *L. pellucidum*, at two locations, where they co-occur and are abundant to understand the geographical variations in shell characters under consideration and further assess their reliability as diagnostic characters.

## MATERIALS AND METHODS

All the *Leptopoma* specimens included in this study were obtained from the *BORNEENSIS* Mollusca collection at the Institute of Tropical Biology and Conservation in Universiti Malaysia Sabah. The collection houses more than 4,000 specimens of *Leptopoma* spp. collected since 2000 from various locations in Sabah (Fig. 1). From this comprehensive collection, 77 alcohol-preserved specimens of four species (*L. sericatum*, *L. pellucidum*, *L. atricapillum*, *L. undatum*) were selected for molecular analysis. 249 empty shells of adult snails of *L. sericatum* (114) and *L. pellucidum* (135) from Balambangan Island and the Kinabatangan region, where both species exist sympatrically were selected for morphological analysis (File S1). These *Leptopoma* specimens were identified as either *L. pellucidum* or *L. sericatum* on the basis of the presence/absence of distinct spiral ridges on the shell (*Vermeulen, 1999*).

### Data collection
#### Genetic data
Genomic DNA was extracted from a total of 77 individuals preserved in 70% ethanol, but sequence data for at least two genes could only generated for 17 of these (Table S1 in File S2). DNA was extracted from foot tissue by using DNeasy extraction kit (Qiagen Inc., Hilden, Germany) according to manufacturer instructions. We used two mitochondrial genes, the protein coding COI (cytochrome oxidase 1) and the non-coding 16S rDNA, and one nuclear gene ITS-1 (internal transcribed spacer 1). Universal primers LCO1490 (5′-GGTCAACAAATCATAAAGATATTGG-3′) and HCO2198 (5′-TAAACTTCAGGGTGACCAAAAAATCA-3′) were used to amplify and sequence COI (*Folmer et al., 1994*). The 16S rDNA region was amplified using primers 16Sar (5′-CGCCTGTTTATCAAAAACAT-3′) and 16Sbr (5′-CCGGTCTGAACTCAGATCACGT-3′) (*Kessing et al., 1989*). ITS-1 region was PCR-amplified using the primers 5.8c (5′-GTGCGTTCGAAATGTCGATGTTCAA-3′) and 18d (5′-CACACCGCCCGTCGCTA
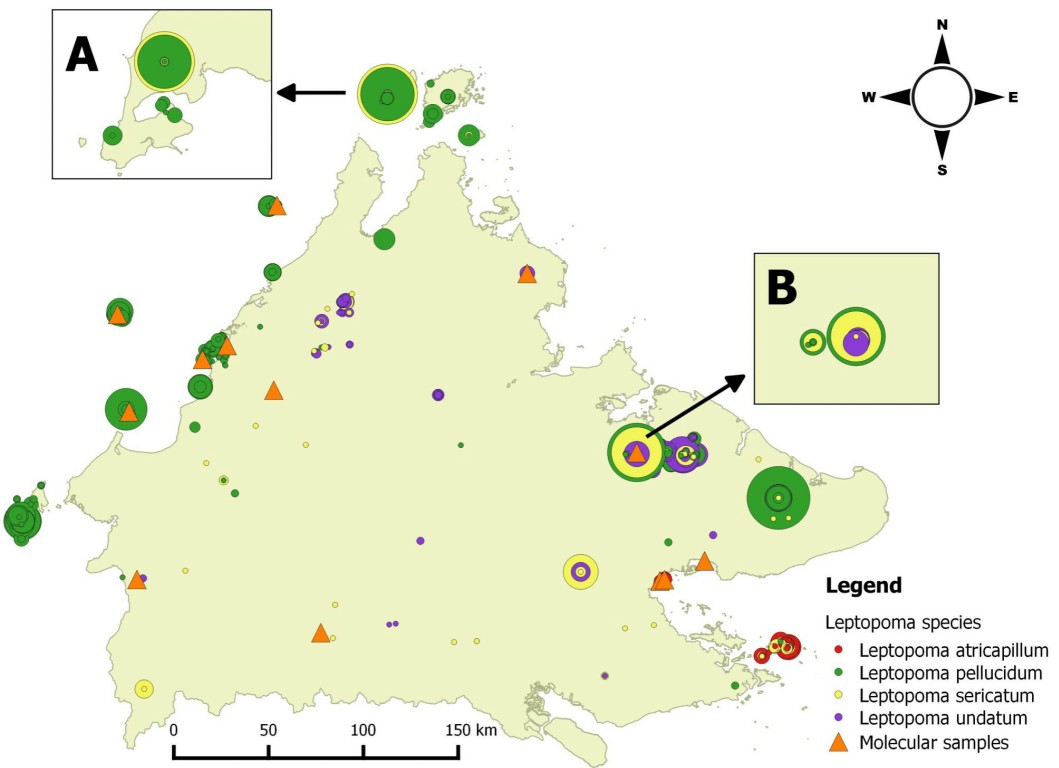

**Figure 1** **Distribution map of four *Leptopoma* species in Sabah based on the records from *BORNEEN-SIS* Mollusca collection, Universiti Malaysia Sabah and localities of molecular samples.** Each circle represents a collection lot for a single *Leptopoma* species with the size of circle indicating the number of specimens in the lot (smallest circle, 1 individuals; largest circles, 140 individuals). The insets (A) and (B) show the sympatric populations of *L. sericatum* and *L. pellucidum* (i.e., on Balambangan Island and in Kinabatangan), which were used for shell morphological analysis. Orange triangles show localities of specimens for molecular study.

CTACCGATTG-3′) (*Hillis & Dixon, 1991*). Thermal cycling was performed with pre-denaturation at 90 °C for 2 min, denaturation at 94 °C for 45 s, one minute of annealing at 55 °C, 60 °C, and 54 °C for COI, 16S rDNA, and ITS-1 respectively, extension at 72 °C for one minute followed by final extension at 72 °C for 5 min. The denaturation, annealing and extension steps were repeated for 35 cycles. The PCR products were sequenced at Macrogen, Inc. (Korea). All sequences were subsequently uploaded and stored in Barcoding of Life Database (BOLD, http://www.boldsystems.org, (*Ratnasingham & Hebert, 2007*)), under the project title "*Leptopoma* in Sabah" (Code: LEPT).

### Shell morphological characters data

Shell form in this study included both quantitative (i.e., size and shape) and qualitative (i.e., colour patterns, spiral ridges and presence/absence dark apertural band) shell characters. These morphological characters were evaluated from the apertural view of the 264 shells examined in total (i.e., the 249 dry specimens and 14 adult specimens included in the phylogenetic analysis). First, high quality photographs were taken of the aperture of each
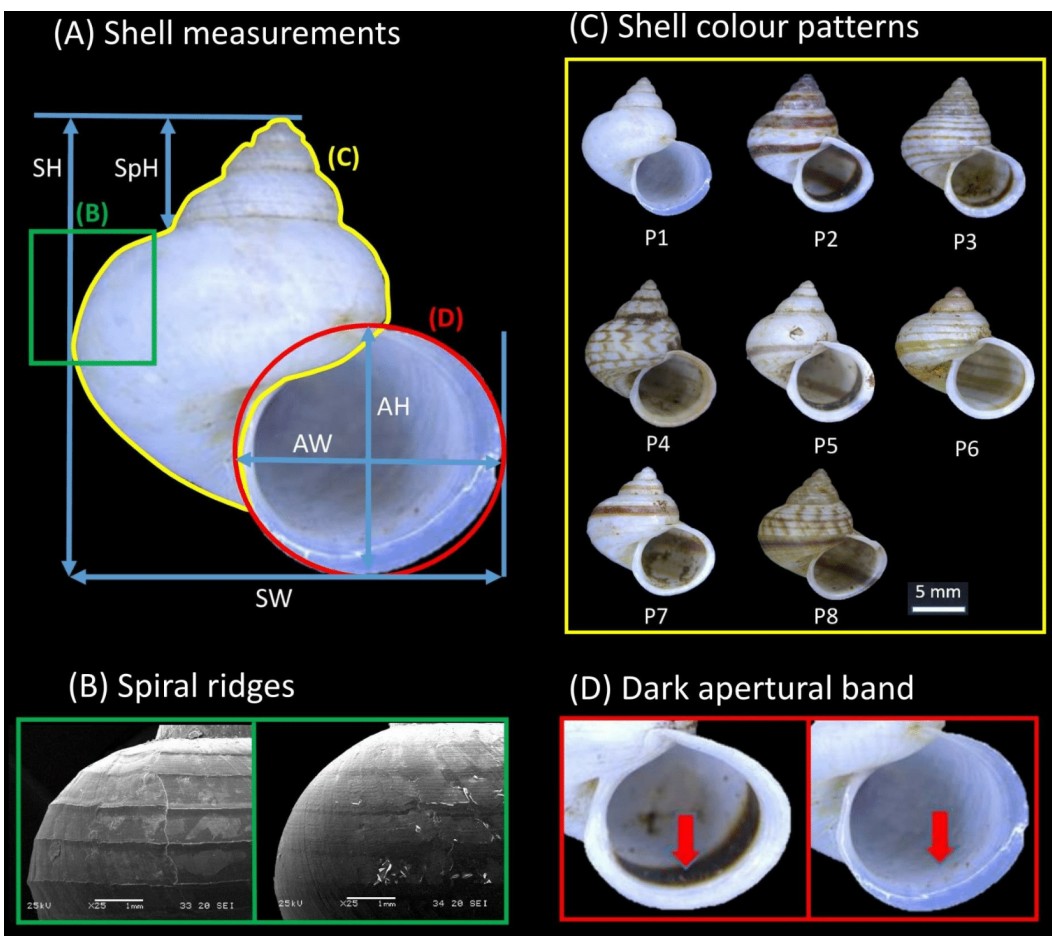

**Figure 2 Qualitative and quantitative shell characters included in the study were assessed on the basis of the shell apertural view.** (A) The five shell quantitative measurements: SH, Shell height; SW, Shell width; AH, Aperture height, AW, Aperture width; SpH, Shell spire height. (B) Spiral ridges: Left—Strong, Right—Weak. (C) The eight distinct shell colour patterns identified in the study. (D) Dark apertural band: Left—Presence, Right—Absence.

shell with the aid of a Leica Stereo Microscope M205. Five quantitative linear measurements, namely shell height (SH), shell width (SW), aperture height (AH), aperture width (AW), and shell spire height (SpH) were then measured directly from the photographs using Leica Application Suite software (Fig. 2A). A sixth quantitative shell character—the ratio of shell height to width—was computed. Next, the states for two qualitative shell characters (i.e., eight distinct shell colour patterns and presence/absence of the dark apertural band) were recorded for each of the shells (Figs. 2C and 2D; see Table S2 in File S2 for the descriptions of the eight shell colour patterns).

## Data analysis
### *Molecular phylogenetic analysis*
In addition to the sequences collected from 17 specimens in this study, 16S rDNA and COI sequences of *L. tigris*, *L. vitreum* and an outgroup species—*Cyclophorus formosensis*

(Nevill, 1882) from *Lee, Lue & Wu (2008a)* and *Nantarat et al. (2014a)*—were obtained from GenBank (File S1, Page 1: Table S1 for informations of specimens). All the DNA sequences were aligned and checked manually using Bioedit v7.1.9 (*Hall, 1999*). In order to find the best-fit model of substitution, jModelTest2 (*Darriba et al., 2012*) as implemented in CIPRES portal (*Miller, Pfeiffer & Schwartz, 2010*) was performed based on corrected Akaike Information Criterion (AICc) for ITS-1 sequences, 16S rDNA sequences and each of the codon positions of COI sequences. Phylogenetic trees were estimated using Maximum likelihood (ML) and Bayesian Inference methods (BI) as implemented in CIPRES portal (*Miller, Pfeiffer & Schwartz, 2010*). ML analysis was conducted using Raxml-HPC2 (*Stamatakis, 2014*) with 1,000 rapid bootstraps. BI analysis was performed using MrBayes v3.2.3 (*Huelsenbeck et al., 2001*). This consisted of running four simultaneous chains for 100,000 generations and 10 sampling frequency. The first 250 trees were discarded as burn-in, while the rest were used to obtain the final consensus tree.

### Phylogenetic signal analysis

Phylogenetic signal analysis was used to investigate the relationship between phylogeny and morphology, with all the analyses done in the R statistical environment, version 3.1.3 (*R Core Team, 2015*). The tips in the tree corresponding to juvenile specimens and outgroup taxa were excluded by using package 'ape' (*Paradis, Claude & Strimmer, 2004*). The final tree for phylogenetic signal analysis consisted of 14 adults of the three *Leptopoma* species, *L. atricapillum*, *L. pellucidum*, and *L. sericatum*. The six quantitative and three qualitative shell characters were mapped onto the tree by utilising package 'phytools'. Phylogenetic signals for each of these nine shell characters were examined using maximum likelihood ($\lambda$) (*Pagel, 1999*) and $K$ (*Blomberg, Garland & Ives, 2003*). The consensus tree was transformed into an ultrametric tree after which a lambda analysis was performed using the 'chronopl' function from the 'ape' package (*Paradis, Claude & Strimmer, 2004*). As a result, a chronogram was generated using penalised likelihood with an arbitrary lambda value of 0.1, the alternative model. A null model, the *Leptopoma* phylogenetic tree with $\lambda = 0$ (no phylogenetic signal), was generated using the 'rescale' function from the 'geiger' package (*Harmon et al., 2008*). The $\lambda$ value of each shell character was estimated for both models using the 'fitDiscrete' function for three qualitative shell characters and 'fitContinuous' function for six quantitative shell characters in the 'geiger' package (*Harmon et al., 2008*). Likelihood scores for the alternative and null models were compared by performing a likelihood ratio test in order to examine the phylogenetic signal in each shell character, wherein Blomberg's $K$ was calculated using the 'physig' function from the 'phytool' package ((*Revell, 2012*); R script in File S3).

### Geographical variation in shell morphology analysis

Two-way ANOVA tests were performed to determine if there were differences in the six quantitative shell characters between: (i) the two *Leptopoma* species (*L. pellucidum* and *L. sericatum*), and (ii) the two locations (Balambangan Island and Kinabatangan). In addition, the interaction effects of both factors (species and location) were tested. A Shapiro–Wilk test for normality (*Shapiro & Wilk, 1965*), and a Levene's test (*Brown & Forsythe, 1974*) for homogeneity of variance, revealed that some datasets were not normally

distributed and showed non-homogeneity of variances (Tables S3 and S4 in File S2). Nevertheless, two-way ANOVA tests were still conducted since the deviations of these datasets from the ANOVA assumption were considered not too serious, and the ANOVA was considered a robust test against the normality assumption (*Zar, 1999*).

Chi-square two-way contingency table tests were performed to determine whether the shell colour patterns and the presence/absence of a dark apertural band were associated with species identity and location respectively. Prior to the analyses, four two-way contingency tables were produced. These summarised frequencies as follows: (1) shell colour patterns *vs.* species, (2) shell colour patterns *vs.* location, (3) dark apertural band *vs.* species, and (4) dark apertural band *vs.* location. Each of the tables was analysed by using Pearson's Chi-squared test. When the expected frequency in the contingency table was less than 5, a Fisher exact test was performed instead of the Pearson's Chi-squared test (*Bower, 2003*). All the statistical analyses were performed in the R statistical environment version 3.1.3 (*R Core Team, 2015*) with the significant p-values set at 0.05. (R script in File S3).

## RESULTS

### The molecular phylogeny of the *Leptopoma* species in Sabah

A total of 660 nucleotide sites were aligned for the COI gene, 558 nucleotide sites for the 16S rDNA gene and 627 nucleotide sites for ITS-1 (File S4). Gaps were treated as missing data and were retained in all phylogenetic analyses. The aligned COI dataset consisted of 36.9% GC content, 207 (31.4%) parsimony informative, and 253 (38.3%) variable sites. The aligned 16S rDNA had 33.3% GC content with 276 (49.8%) parsimony informative and 406 (73.3%) variable sites. On the other hand, aligned ITS-1 gene had 48.6% GC content, 158 (25.2%) parsimony informative, and 274 (43.7%) variable sites. Phylogenetic analyses were run for four datasets: three single-gene datasets (ITS-1, COI, 16S rDNA) and a concatenated dataset of the three genes. The tree was rooted on the outgroup *Cyclophorus formosensis* (Fig. 3).

Best-fitted models were selected based on the corrected Akaike Information Criterion (AICc); the models were TPM3uf+G for ITS-1, TIM3+G for 16S rDNA, TIM3ef+G, TPM3uf+I, and TPM3uf+G respectively for first, second and third codons of COI. These models were used in both ML and BI analyses. Phylogenetic trees from the ML and BI analyses of the concatenated dataset showed no conflict in tree topology. The monophyly of the three *Leptopoma* species (*L. sericatum*, *L. pellucidum* and *L. atricapillum*) in Sabah was consistently strongly supported (posterior probability of 100% and bootstrap support greater than 75).

### Phylogenetic signals in shell characters of the *Leptopoma* species

Figure 4 shows the correlation between phylogeny and the quantitative and qualitative shell characters for *L. sericatum*, *L. atricapillum* and *L. pellucidum*. A phylogenetic signal test based on Pagel's $\lambda$ and Blomberg's $K$ showed that spiral ridges and the presence/absence of dark apertural band represented a strong signal with $\lambda = 1$ and $K > 1$ ($K = 4.536$ for spiral ridges and $K = 1.114$ for dark apertural band) (Table 1). However, shell colour pattern, a character that is often used as a diagnostic character in traditional classification, showed a
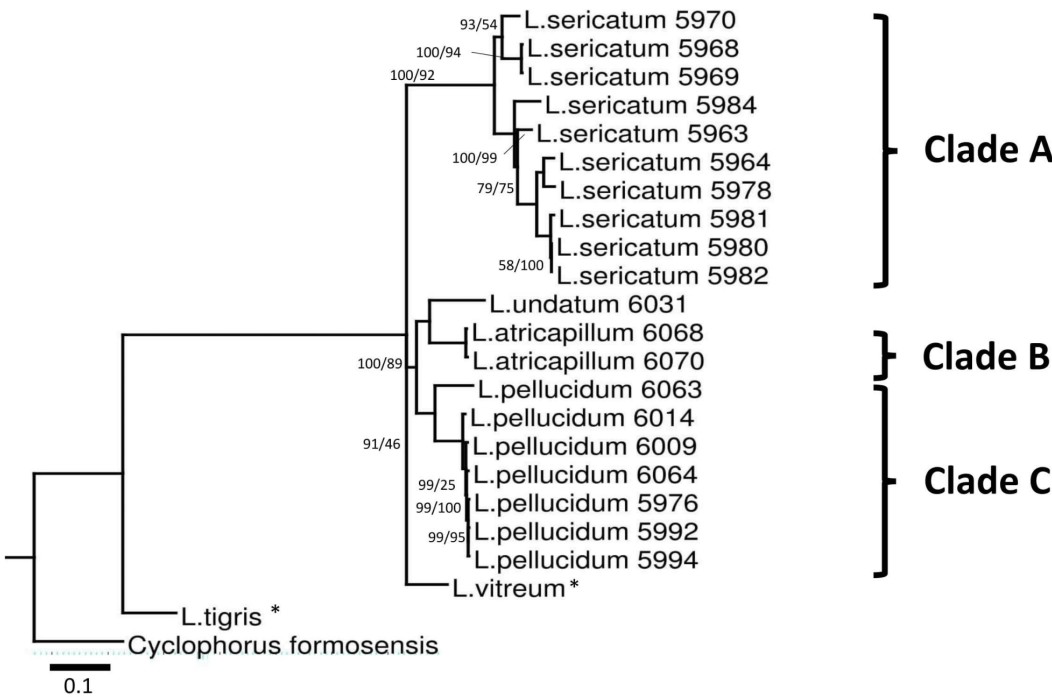

**Figure 3** **Bayesian inference tree of *Leptopoma* spp. based on concatenated dataset of 16S rDNA, COI and ITS-1.** Support values on branches are Bayesian posterior probability (BI) followed by maximum likelihood (ML) bootstrap value. Internal branches with ML bootstrap value = 100% and *PP* value = 100 were not represents in the figure. The number shown beside each specimen of Sabah *Leptopoma* is the relevant specimen number (Table S1 in File S2), and the specimens with asterisk are non-Borneo's *Leptopoma* species (i.e., sequences obtained from Genbank). Clades A, B and C indicate the three Sabah species of focal interest. Scale bar for branch length = 0.1 substitutions per site.

weak phylogenetic signal ($\lambda = 0.997$, $K = 0.234$). Among the quantitative shell characters, shell height exhibited a strong signal according to Pagel's $\lambda$ although Blomberg's $K$ gave a weak phylogenetic signal. The ratio of shell height to width (SH/SW) exhibited the weakest phylogenetic signal among all the shell characters ($\lambda = 0$, $K = 0.054$).

### Geographical variation in shell forms

Two-way ANOVA showed that all shell quantitative characters (except aperture height) differed between the two locations (Table 2, Fig. 5). In addition, all shell quantitative characters except shell width and aperture height also differed between the two species. There was interactive effect of species and location on the aperture height, shell spire height and ratio of shell height to width.

Chi-square analyses indicated a significant association between the frequencies of shell colour patterns and both species identity (Fisher's exact test: $p = 0.0000$) and location (Fisher's exact test: $p = 0.0000$). On the other hand, there was a fairly significant association between the frequency of presence/absence of dark apertural band and species identity (Pearson's Chi-Squared with Yates' continuity correction: $X^2(1, N = 249) = 4.019$, $p = 0.0449$) but not with location (Pearson's Chi-Squared with Yates' continuity correction:

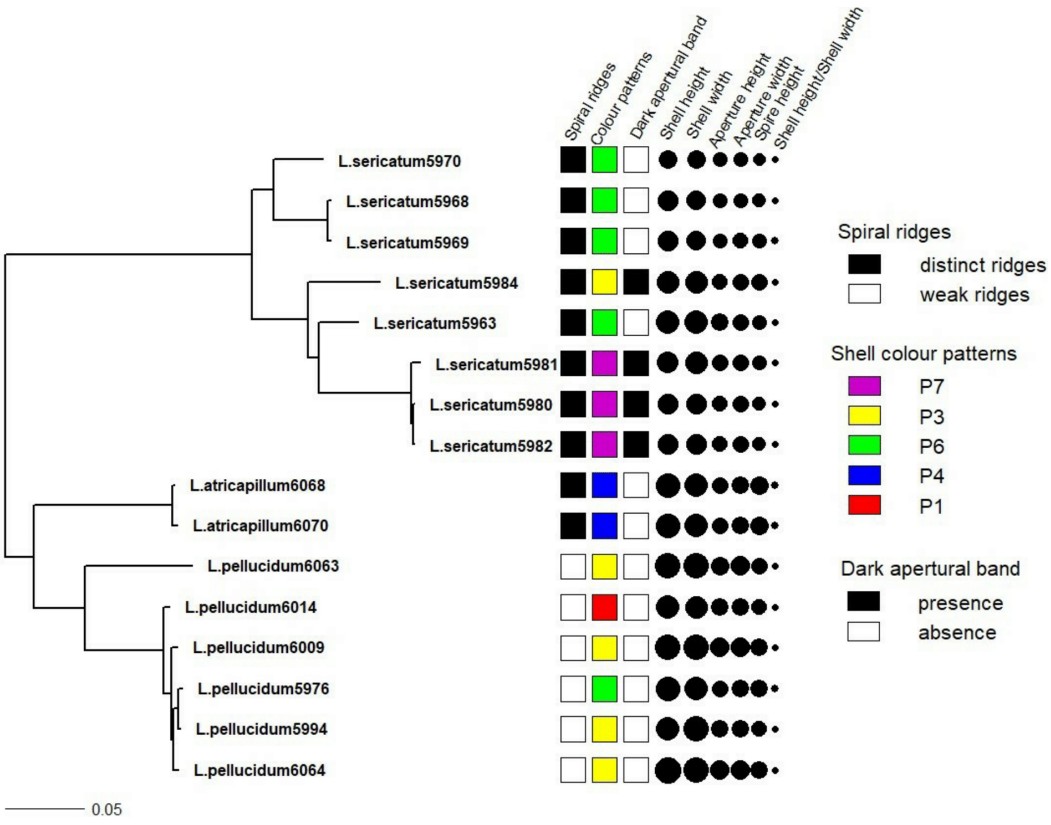

**Figure 4  Shell quantitative and qualitative shell characters as mapped on to the phylogenetic tree.** Tree as in Fig. 3 with the juvenile specimens were dropped from the tree and only the 14 adult specimens of the three *Leptopoma* species were retained. Different categories of the three qualitative shell characters: spiral ridges, shell colour patterns and dark apertural band (Figs. 1A–1C respectively) were represented by different colour of the squares; and the six shell quantitative measurements: shell height, shell width, aperture height, aperture width, shell spire height and ratio of shell height to width were represented by the size of the circle.

$X^2(1, N = 249) = 1.5505, p = 0.2131$). Both contingency tables are available in Tables S5 and S6 in File S2. Overall, the shell characters considered in this study did not show consistent differences between *L. pellucidum* and *L. sericatum* since the differences in shell forms were coupled with geographical variations and interaction effects between geography and species.

## DISCUSSION

This study presents the first molecular phylogeny investigation on genus *Leptopoma* in Sabah, one of the many understudied genera within the Cyclophoridae, and examines the concordance between morphology and phylogeny as well as geographical variation in shell form.

We found consistently significant support for the monophyly of the three morphologically similar Sabah *Leptopoma* species, *L. pellucidum*, *L. sericatum* and
**Table 1** **Phylogenetic signal test result acquired from Pagel's λ method and Blomberg's *K* method.** Values equal to 1 or more than 1 were bolded.

| Shell traits | Lambda (λ) | Likelihood score (alternative model) | Likelihood score (null model, λ = 0) | *p*-value | *K* | *P* |
|---|---|---|---|---|---|---|
| Patterns | 0.997 | −17.986 | −21.906 | 0.005 | 0.234 | 0.014 |
| Spiral ridges | **1.000** | −3.654 | −9.704 | 0.0005 | **4.490** | 0.001 |
| Dark ring band | **1.000** | −4.418 | −7.274 | 0.017 | **1.317** | 0.007 |
| AH | 0.998 | −15.969 | −21.266 | 0.001 | 0.518 | 0.001 |
| AW | 0.866 | −16.641 | −21.395 | 0.002 | 0.437 | 0.001 |
| SpH | 0.894 | −17.850 | −22.426 | 0.002 | 0.444 | 0.003 |
| SH | **1.000** | −24.197 | −29.651 | 0.0007 | 0.567 | 0.001 |
| SW | 0.829 | −24.040 | −29.651 | 0.001 | 0.442 | 0.001 |
| SH/SW | 0 | 17.147 | 17.147 | 1 | 0.056 | 0.320 |

**Notes.**
Abbreviations: SH, shell height; SW, shell width; AH, aperture height; AW, aperture width; SpH, shell spire height; SH/SW, ratio of shell height and width.

**Table 2** **Two-way ANOVA for the effect of geographical variation and species identity on six quantitative shell traits.** Significant *p*-values were bolded.

| | Geographical region | | | Species identity | | | Geographical* Species | | |
|---|---|---|---|---|---|---|---|---|---|
| | *df* | *F* | *P*-value | *df* | *F* | *P*-value | *df* | *F* | *P*-value |
| SH | 1 | 18.88 | **2.03e−05** | 1 | 12.763 | **0.0004** | 1 | 3.551 | 0.0607 |
| SW | 1 | 5.376 | **0.0212** | 1 | 0.104 | 0.7473 | 1 | 0.586 | 0.4447 |
| AH | 1 | 0.086 | 0.770 | 1 | 0.000 | 0.987 | 1 | 16.185 | **7.66e−05** |
| AW | 1 | 4.235 | **0.0407** | 1 | 4.399 | **0.0370** | 1 | 1.994 | 0.1592 |
| SpH | 1 | 24.92 | **1.14e−06** | 1 | 36.33 | **6.08e−09** | 1 | 80.01 | **<2e−16** |
| SH/SW | 1 | 17.36 | **4.29e−05** | 1 | 62.10 | **1.07e−13** | 1 | 5.53 | **0.0195** |

**Notes.**
Abbreviations: SH, shell height; SW, shell width; AH, aperture height; AW, aperture width; SpH, shell spire height; SH/SW, ratio of shell height and width..

*L. atricapillum*. This finding is in concordance with the existing traditional morphology-based classification. For example, take the placement of *Leptopoma pellucidum* 6014 (Fig. 3) in our phylogeny, this population was previously assumed to be *L. vitreum* (*Duperrey, Lesson & Garnot, 1830*) due to its uniformly white shell, but we have shown that it falls within the *L. pellucidum* clade, and this provides support to *Vermeulen (1999)*'s decision to assign *L. vitreum* from Sabah to *L. pellucidum*. In the case of *L. pellucidum* and *L. sericatum*, the presence of intermediate forms between the two species led Vermeulen to recognise them as being distinct on a purely provisional basis. In this study, results suggested that the two species could be unequivocally regarded as separate. In short, the findings of this study are in line with past research in this region which proposed that a combination of morphology and molecular approaches could improve taxonomy of land snails (*Nantarat et al., 2014a*; *Nantarat et al., 2014b*; *Liew, Schilthuizen & Vermeulen, 2009*; *Liew et al., 2014*). However, we aware that this findings still need to be verified with more specimens of both species from their entire distribution range outside of Sabah.

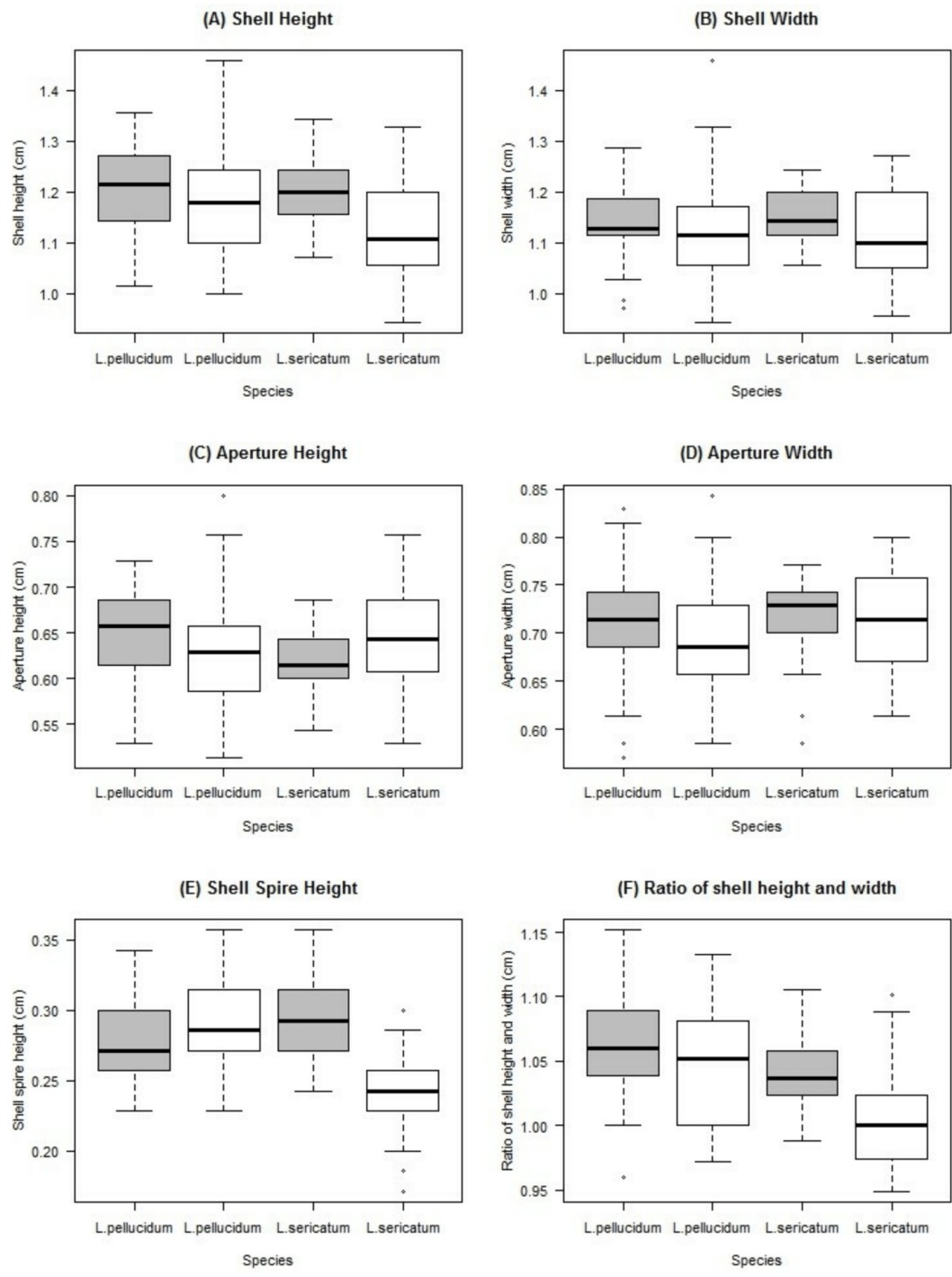

**Figure 5** **Boxplots show the differences of the six quantitative measurements of shell for the** ***Leptopoma pellucidum*** **and** ***L. sericatum*** **in each of the two locations (Balambangan Island and Kinabatangan region).** Grey boxplot indicated sample from Balambangan Island (BI) and white boxplot indicated sample from Kinabatangan ($K$). Sample sizes for each dataset were: BI-pellucidum ($n = 45$); K-pellucidum ($n = 90$); BI-sericatum ($n = 46$); K-sericatum ($n = 68$).

A morphological character is assumed to have strong phylogenetic signal when the same character clusters together within closely-related species (*Blomberg, Garland & Ives, 2003*). This could be a useful diagnostic indicator for species delimitation. The phylogenetic signal tests showed that spiral ridges had a significant phylogenetic signal ($\lambda = 1$, $K > 1$). Distinct spiral ridges were present in *L. sericatum* and *L. atricapillum,* while *L. pellucidum* had weak spiral ridges. This indicated that weak spiral ridges might be an automorphy character for *L. pellucidum* which could be useful in discriminating *L. pellucidum* from *L. sericatum* and *L. atricapillum*. This result was in agreement with *Vermeulen*'s (*1999*) work where spiral ridges were also used as a key to delimit *L. pellucidum* and *L. sericatum*.

The presence of a dark apertural band has not been observed in other cyclophorids and was not mentioned in other revisionary works of *Leptopoma* species. Results showed that the presence/absence of a dark apertural band exhibited a significant phylogenetic signal and does not associate with geography for two of the species. Nevertheless, association of dark apertural band with species identity was fairly significant. Our results showed that dark apertural band only present in some *L. sericatum,* with this character being observed at just two sites (i.e., Kinabatangan and the Tabin Wildlife Reserve area). However, when observing all the specimens in the *BORNEENSIS* collection, we found that such character is actually present in both *L. pellucidum* and *L. sericatum*. The apertural band of *L. pellucidum* was darker than *L. sericatum*. We also observed that dark apertural band is generally present in shells with thickened outer lip (gerontic shell) and a high abundance of shells with such character can be found in the Tabin Wildlife Reserve area. This might indicate a longer life span of *Leptopoma* species in the area, though this requires proper investigation. Overall, our findings here suggest that although this character shows a strong phylogenetic signal, it is not an appropriate character for species-level identification in *Leptopoma*.

Shell colour patterns are usually used as one of the key determinants to discriminate between species in traditional taxonomic classification. One of the sister taxa of *Leptopoma*, the species in genus *Cyclophorus*, was distinguished unambiguously based on shell patterns that were also supported by molecular data (*Nantarat et al., 2014b*). Compared to genus *Cyclophorus*, shell colour patterns of the *L. sericatum* and *L. pellucidum*, exhibited a weak phylogenetic signal. This case of shell colour pattern polymorphisms of the two *Leptopoma* species is similar to other well-known land snails such as *Cepaea nemoralis* and *C. hortensis* (*Owen & Bengtson, 1972*; *Ożgo & Schilthuizen, 2012*; *Cameron & Cook, 2012*; *Cameron, 2013*). However, unlike *Cepaea* land snails that have been studied extensively, the causal mechanism for the *Leptopoma* land snail's diverse shell colour patterns is still unknown. This study also revealed that the *Leptopoma* species in Sabah exhibits idiosyncratic differences between locations in the degree of shell colour patterns polymorphism. For example, the *Leptopoma* population at Balambangan Island has more shell colour patterns as compared to the population at Kinabatangan. As a result, the geographically-associated variations in shell colour patterns and weak phylogenetic signal strongly indicate that this character is unreliable as a diagnostic character for the identification of the *Leptopoma* species considered in this study.

Significant inter- and intra-specific variations in quantitative shell characters have been noted both in family Cyclophoridae (*Lee, Lue & Wu, 2012*; *Nantarat et al., 2014b*) and in

 

other gastropods (*Kameda, Kawakita & Kato, 2007*; *Desouky & Busais, 2012*; *Hirano et al., 2014*). In *Vermeulen*'s (*1999*) descriptions of *L. sericatum* and *L. pellucidum*, the ratio of shell height to width of *L. sericatum* is slightly smaller than *L. pellucidum*. However, intermediate in ratio of shell height to width between the two species occur and lead to weak phylogenetic signal in this character. From the phylogenetic signal test, only shell height produced a significant signal. Nevertheless, this study revealed a high degree of geographical variations in the quantitative shell characters; for example, both *Leptopoma* species from Balambangan Island were larger than the same species found in Kinabatangan. Previous studies suggested that land snails found on islands tend to undergo extensive morphological diversification (*Johnson & Black, 2000*; *Stankowski, 2011*). In view of this, quantitative shell characters are thus not advisable as a diagnostic indicator to delimit among these two *Leptopoma* species in Sabah due to the strong influence of geographical variations.

While our findings reflected the reliability of 'hard' character (i.e., shell sculpture) over colour patterns and banding, and revealed considerable geographical variation in some shell characters, the findings from this study were based exclusively on *Leptopoma* species from Sabah. For future work to improve the taxonomy of this genus, the study needs to be extended to include larger numbers of specimens from a larger geographical area. In addition, more genetic markers and examination of reproductive characters are required to elucidate comprehensive phylogeny and morphological variation among the *Leptopoma* species

## CONCLUSION

This study has revealed partial information on the phylogeny and morphology variations of all *Leptopoma* species in their entire distribution range. Despite its small geographical scale, the study has resolved taxonomic uncertainties of three *Leptopoma* species in Sabah and revealed notable variations in both the quantitative and qualitative shell characters for the species. From the findings, it is suggest that any future revisionary attempt of the taxonomy on the rest of *ca.* 100 *Leptopoma* species should consider the possible caveats in using the shell characters as the sole evidence and should include molecular markers in the study. Further studies that include more samples from a wider geographical reach are strongly recommended.

## ACKNOWLEDGEMENTS

We would like to thank Cornelius Peter for his help with molecular lab work and Joumin Rangkasan who assisted with fieldwork. This project was carried out by CCP and PSH as part of their undergraduate honours degree dissertation. We thank Rudiger Bieler, Siong Kiat Tan and an anonymous reviewer for their constructive comments.

### Funding

This research is supported by Universiti Malaysia Sabah through grant no. SLB0107-STWN-2015 to Thor-Seng Liew and MyBrainSc scholarship to Chee-Chean Phung. The funders had no role in study design, data collection and analysis, decision to publish, or preparation of the manuscript.

### Grant Disclosures

The following grant information was disclosed by the authors:
Universiti Malaysia Sabah: SLB0107-STWN-2015.
MyBrainSc scholarship to Chee-Chean Phung.

### Competing Interests

The authors declare there are no competing interests.

### Author Contributions

- Chee-Chean Phung conceived and designed the experiments, performed the experiments, analyzed the data, contributed reagents/materials/analysis tools, wrote the paper, prepared figures and/or tables, reviewed drafts of the paper.
- Pooi-San Heng conceived and designed the experiments, performed the experiments, analyzed the data, contributed reagents/materials/analysis tools.
- Thor-Seng Liew conceived and designed the experiments, analyzed the data, contributed reagents/materials/analysis tools, wrote the paper, prepared figures and/or tables, reviewed drafts of the paper.

### DNA Deposition

The following information was supplied regarding the deposition of DNA sequences:
Genbank—see File S2, Table S1.

### Data Availability

Liew, Thor Seng; Phung, Chee-Chean (2017): Land snails of Leptopoma Pfeiffer, 1847 in Sabah, Northern Borneo (Caenogastropoda: Cyclophoridae): an analysis of molecular phylogeny and geographical variations in shell form. figshare.
https://doi.org/10.6084/m9.figshare.5165632.v2.

### Supplemental Information

Supplemental information for this article can be found online at http://dx.doi.org/10.7717/peerj.3981#supplemental-information.

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
