# Peer review of "Land snails of Leptopoma Pfeiffer, 1847 in Sabah, Northern Borneo (Caenogastropoda: Cyclophoridae): an analysis of molecular phylogeny and geographical variations in shell form"

_PeerJ, doi:10.7717/peerj.3981_

## Round 0.1 · original submission · Minor Revisions

Two expert reviewers have commented on your interesting manuscript and, as you will see, have left comments and annotations that amount to a series of requested minor revisions. I agree with the reviewers and only have a few additional comments and requests:

Please capitalize formal taxonomic names above the species level (e.g., Mollusca, not mollusca), but not informal usages (e.g., cyclophorids, not Cyclophorids). I suggest using "fluid-preserved" or "alcohol-preserved" instead of referring to "wet" specimens. In which context are you referring to Cyclophoridae as the "most diverse family" (line 252) -- as land snails, caenogastropods, gastropods, or Mollusca? Also, I assume you mean "morphologically ill-defined" where you write "morphologically-ill" (line 256)?

I look forward to receiving the revised manuscript.

Reviewer 1 ·

Basic reporting

This preliminary study of Leptopoma from Sabah is interesting and important because:
a) because studies on the molecular systematics of Cyclophoroideans and the Familt Cyclophoridae are few and far between and this study is the first of its kind for the genus Leptopoma and
b) because it seeks to explore the utility of traditional shell-based taxonomic characters in a quantitative way.

Overall it meets all 5 of the basic reporting requirements, but I recommend that the following changes should be made:
- The sampling for the molecular component of their study is limited so they need to state in the discussion that this is a preliminary study looking at a small individuals at a small number of sites and adapt their discussion accordingly
- e.g. clearly highlight the localities sampled in the phylogeny on the map in Figure 1 so that readers can differentiate those localities from all other sites.
- Avoid repetition – don’t repeat what has been said in the introduction in the discussion and conclusions
- Standardise and explain/refine terminology and use it in a consistent way.
- e.g. Explain what is meant by shell form right at the beginning of the article by stating that it includes all shell morphological characters (shape, colour pattern, ornamentation). Actually, I would avoid using the word ‘form’ in this context because it often implies shape/size or colour pattern. Instead I would use shell morphology here and use form in the context of the 8 colour patterns - you can refer to these as colour forms. In this way you avoid confusion with your other use of shell colour pattern (i.e. one of the 3 characters used to define the 8 colour forms)
- e.g. replace the term ‘apertural ring band’ with ‘apertural ring’ or ‘apertural band’
- e.g. Don’t use alternative terms such as characters/traits – stick to one option – seeing as this is a taxonomic paper I would use characters and drop the use of traits.

Experimental design

Overall it meets all 5 of the experimental design requirements

Obviously a lot of thought has gone into the experimental design and an attempt has been made to use some very recently developed tools/ approaches – that’s very commendable as is the effort that has been made to explore morphological variation in a quantitative way – the authors are really trying to tackle a difficult problem boldly. That said, they shouldn’t overemphasise the broader relevance of their findings, but more on this below.

Validity of the findings

While the authors’ data/results do support their conclusions, they shouldn’t overemphasise the generality of their results. There are three issues here:
1) Having only sampled 4 of the c. 100 species in the genus they shouldn’t conclude that their results are generally applicable to the whole genus. Clearly their findings indicate that there may be a wider problem, but this issue has still to be investigated.
2) Nowhere is it made clear what the global ranges of the 4 Leptoma species. Are these species endemic to Sabah? Or are these species more widely distributed? I think it is important to provide a little information on this in the methods or even in the intro. e.g. Species X has a range encompassing the Malay Peninsula, and the islands of Borneo and Sumatra. Provide references as appropriate/if available. If most of what is currently known is largely based on the BORNEENSIS dataset the authors should make this clear.
3) As with 2), it is not clear how representative this study is in terms of the overall species richness of Leptopoma in Sabah. What proportion of Sabah’s Leptoma species are included in this study?

In the discussion the authors need to briefly lay out what they would do to improve the design so that they can improve their understanding of the Sabah Leptopoma and broaden the relevance of their findings e.g. improved sampling from across the ranges of the 4 species, improved population sampling.

Additional comments

I have made lots of suggestions directly on the pdf of the MS - please have alook at these. Please also re-write the abstract to reflect the changes I have suggested. Make it clear in the abstract that the study is based on a new/recently gnerated datatset, that your study deals with the taxonomy of 3 species in Sabah and that your findings necessarly reflect this - i.e. avoid extrapolating from your study to the whole genus.

I look forwad to seeing more from the authors on this very interesting and intriguing system and it is great that they are trying to quantify shell characters even though it is an extremely challenging thing to do.

Annotated reviews are not available for download in order to protect the identity of reviewers who chose to remain anonymous.

·

Basic reporting

Species delimitation is known to be problematic in the genus Leptopoma, which is traditionally classified based solely on shell morphology. This is an interesting study to investigate the molecular phylogeny of similar-looking species in Sabah, and examines the reliability and suitability of some conchological characters for species delimitation in two polymorphic species that occur sympatrically. The figures and tables are clear, and as a whole, the manuscript is well structured. Nevertheless a slight change in the title is suggested because there is insufficient evidence to suggest that the variations are due to geography.

Experimental design

Adequate and methodical.

Validity of the findings

The findings are sound. The unreliability of certain characters, notably shell colour and patterns, used in shell morphology based taxonomy are well elucidated by the results.

Additional comments

I suggest that the introduction include a little explanation regarding the presence of a “dark ring band” in the aperture of some specimens of both Leptopoma pellucidum and L. sericatum. It is not apparent whether this character is only found in the Kinabatangan population and nowhere else. Is this found only in gerontic shells with a much thickened outer lip? As the authors have mentioned in the discussion, this interesting character has not been noted in other works. Hence some details will be informative.

The problem of species delimitation with intermediate shell forms has not been properly treated? Were intermediates sequenced or included in this study? I would be interested to know the status of these intermediates.

Most experienced taxonomists are well aware of the limitations of using shell colour and patterns for species identification. For example many fossils and subfossils, especially of extant species, without discernible colour can usually be quite easily identified by the relevant experts. Students however, often use colour and patterns in their initial attempts at classification. Perhaps a sentence could be added in the conclusion to emphasize the reliability of the ‘hard’ characters such as shell surface sculpture, spire profile, and shape, over colour and patterns.

Overall, the MS is clear and well written. However the language still requires a bit of attention. This is nevertheless a very minor issue and does not affect the quality of the study. I have made some slight edits on the marked PDF attached.

---

## Round 0.2 · Minor Revisions

Thank you for returning your revised manuscript and for providing such detailed comments. I will be glad to recommend this manuscript for publication. Please have a look at my annotations in the attached version -- your new text included a few sentences for which I want to make sure I interpreted your meaning correctly. All my comments are minor and mostly attempt to clarify language issues. Please return your final version at your earliest convenience (no need for a new "rebuttal" letter - just use of my notes what you find helpful) and I shall process this manuscript very quickly.

My apologies for the recent delay. Our field laboratory in Florida got caught up in the recent hurricane that caused much destruction in the region, and I was temporarily out of communication.

---

## Round 0.3 · accepted · Accept

I concur with the adjustments you have made to the text of the manuscript and look forward to seeing this in print!